# Is Satisfaction with Online Learning Related to Depression, Anxiety, and Insomnia Symptoms? A Cross-Sectional Study on Medical Undergraduates in Romania

**Claudiu Gabriel Ionescu** [1,*] , **Anca Chendea** [2] **and Monica Licu** [1]

[1] Department of Medical Ethics, Faculty of Medicine, "Carol Davila" University of Medicine and Pharmacy, 050474 Bucharest, Romania
[2] Department of Psychiatry, Faculty of Medicine, "Carol Davila" University of Medicine and Pharmacy, 050474 Bucharest, Romania
* Correspondence: claudiu.ionescu@drd.umfcd.ro

**Abstract:** The objective of this study was to investigate online learning satisfaction in a sample of university students and its relationship with depression, anxiety, insomnia, and the average number of hours spent online. A total of 463 medical students were recruited for an online survey conducted from February to March 2022 with the main objective of estimating online learning satisfaction, while secondary outcomes involved assessing the relationship between online learning and depression, anxiety, insomnia, and the average number of hours spent online. A total of 285 participants were female (71.4%) and the mean age was 20.2 years. The results revealed that depression, anxiety, and insomnia are negatively correlated with overall satisfaction with e-learning. The more time students spent online, the greater the overall satisfaction. There are significant differences regarding student perceptions of interactivity in online learning satisfaction outcomes ($p < 0.05$, $\eta^2$ partial Eta Squared-0.284). The opportunity to learn via chat-box presented differences in overall satisfaction while pleasant aspects of online learning, such as "no travel" and "economy", were related to satisfaction. The students revealed that the higher the psychopathology scores, the less satisfied they were with online learning, while a higher number of hours spent online contributed positively to satisfaction.

**Keywords:** medical education; e-learning; distance learning; psychopathology; teaching satisfaction; undergraduate students; COVID-19

## 1. Introduction

In March 2020, the World Health Organization (WHO) declared COVID-19 a global pandemic [1–3]. Among the measures implemented to mitigate the pandemic in Romania was the cancellation of in-person teaching and the switch to online learning at universities. Online education is defined as learning through a primarily electronic medium with the interaction between students and their educational materials and activities taking place in a full virtual environment [4]. University closures, online teaching, and the inability to complete hospital internships changed the inherent training pattern for medical students [5]. This included the introduction of new tools for delivering online and hybrid teaching [6,7], but also teleconferences, webinars, and tutorials. These tools have important advantages, allowing students to attend educational sessions from any location and helping hospitals to deliver a variety of educational sessions via reports, case discussions, or live quizzes [8–10]. Whether the students adjusted to the new educational paradigm has been debated controversially and rests on a small number of studies that reported conflicting results [11,12].

The satisfaction with online learning in medical students is based on specific community, educational, social, cultural, and economic differences [13]. The overall results highlight a moderate individual satisfaction and a low perception of effectiveness [14,15]; however, learning outcomes based on academic results, building skills, and interactivity remain unclear [16].

Medical students are known to have a higher psychopathology regarding academic and emotional requirements [17–25]. In this context, our research aim was to perform a more complex investigation into the relationship between online learning satisfaction and psychopathology in medical students with interactivity and average hours spent online used as valuable indicators.

*Recent Literature*

The literature on the effects of COVID-19 on the mental health of medical students reported that the prevalence of depression was higher than in the general population and in healthcare workers, with feelings of vulnerability and loneliness increasing [26,27]. Over 50% of students were experiencing increased stress and anxiety, irregular sleeping patterns, and changes in eating habits brought on by the pandemic, while feeling less productive and displaying an inability to focus and study [28]. The number of hours spent online sky-rocketed during the pandemic; smartphones were used more often for emotional support, increasing the probability of over-use. This is higher in medical students than any other student group and is predicted by anxiety and depression scores and negatively correlated with online learning satisfaction [29–32]. Known factors for worse mental health among medical students include the lack of time and conditions for study, having most extracurricular activities being canceled with no remote alternative, lack of motivation to learn, excessive self-pressure for good grades, and lack of leisure time as a result of the pandemic [33]. To the best of our knowledge, a single study addressed the relationship between satisfaction, depression, and anxiety which reported negative correlations [34].

Taking these data into account, no research to date has addressed the indicators of average hours spent online, insomnia, and online interactivity and their relationship with online learning satisfaction and self-reported psychopathology.

Therefore, there is a need for a detailed understanding of the impact of online learning satisfaction on students' psychopathology which can provide important insights for education professionals and mental health researchers. The overall satisfaction with online learning compared with the classical variant of learning should take into consideration not only the external variables, such as reducing commute costs, saving travel time, flexibility, freedom of action of students, and preventing academic tardiness, as seen in previous studies, but also individual, detailed, self-reported psychopathology, as we aimed to achieve in our study [35]. Some of these variables have already been described in very recent literature, but without emphasis on psychopathology [36–38]. This should be approached as a new research field with emphasis on extensive research, especially for the component "interactivity" and for the potential protective role of digital familiarity translated into hours spent studying online, among others.

We conducted the research as follows. We investigated the bidirectional association between psychopathology and the average number of hours spent online and their level of satisfaction with online learning. We aimed to understand how psychopathology indicators and the average number of hours spent online might be independent predictors for online learning satisfaction. We then explored the positive and negative aspects of online learning, the perception of interactivity, and the overall attitude towards a successful online replacement of classic learning and its relationship with psychopathology, comparing different groups of students based on their answers regarding interactivity.

All hypotheses were summarized in the assertion that all the above variables were interrelated, but no presumptions as to whether the associations were negative or positive were made in advance.

## 2. Materials and Methods

### 2.1. Design

This study used a cross-sectional design with 1 assessment taken between February and March 2022. For this study, the data were extracted through the administration of an online set of questions sent to 1529 first-year medical students containing the study

instruments. We targeted our research towards this specific sample because among all the medical students we had access to at our university (all 3 years of study), this group were exposed the most to online learning at key points in their education (end of high school and first semester attending medical school). The subjects studied by them in their first semester were anatomy, physiology, cell biology, medical psychology, biochemistry, biophysics, and medical marketing. The language used was English because this was the overall orally expressed preference of the Romanian students and the significant amount of English module students (almost 30%). To launch the research, we sent a preliminary email to all first-year students, informing them of the study's aims and soliciting their participation. Participation was anonymous, and responses did not affect the teacher's evaluation of students' performance. The study procedure was run via a SurveyMonkey® interface (One Curiosity Way, San Mateo, CA, USA).

### 2.2. Participants

The study included undergraduate students undergoing their training at the Carol Davila University of Medicine and Pharmacy in Bucharest who were invited to participate via online classroom message and orally at the seminars and courses. The inclusion criteria were (1) being at least 19 years of age, (2) being a current undergraduate student at the above-mentioned institution, and (3) having participated in at least 2 semestrial courses in in-person, online, and hybrid formats.

### 2.3. Context and Procedure

At the time of the study, online learning platforms had already been used for 2 years by teachers and for 1 semester by the first-year students involved in the study. Students were taught using Google Meet® or Zoom® with a strictly fixed schedule resembling the typical classroom schedule, where they had to log in at certain times to attend live lectures and active learning and where they were supposed to turn on their cameras. In fall 2021, a hybrid educational system was implemented. Hybrid teaching comprised of online classes (lectures) and face-to-face classes (seminars, clinical internships) in compliance with prevention measures (e.g., mask mandates).

Procedures in the study were designed in accordance with the World Medical Association Declaration of Helsinki. The study protocol was approved by the Carol Davila University of Medicine and Pharmacy Institutional Review Board (no. 10908/2022).

### 2.4. Instruments

All participants were assessed for symptoms of depression, anxiety, insomnia, and satisfaction with online learning.

(1) Symptoms of depression were assessed with the patient health questionnaire (PHQ-9). The PHQ-9 is a 9-item self-reported scale that assesses the severity of several depressive symptoms over the past 2 weeks. Each symptom is rated on a Likert-scale ranging from 0 (not at all) to 3 (nearly every day) and then summed to a total score ranging from 0–27. The total score ranges from 0 to 27 and is subdivided into 5 categories: 1 to 4 is minimal, 5 to 9 is mild, 10 to 14 is moderate, 15 to 19 is moderate–severe, and 20 and above is severe. Cronbach's alpha in this study was 0.83 [39–41].

(2) Generalized anxiety disorder-7 (GAD-7) is a 7-item questionnaire asking participants how often they were bothered by each symptom, such as feeling nervous, trouble relaxing, irritable, and afraid that something awful might happen, during the last 2 weeks. The GAD-7 has also been identified as a screener for panic disorder, social phobia, and PTSD (Cronbach $\alpha$ = 0.92) [42,43].

(3) The insomnia severity index (ISI) is a 7-item self-report questionnaire measuring subjective sleep difficulties. Items are rated on a 5-point response scale from 0 to 4 with higher scores corresponding to greater symptom severity, and their sum yields a global score ranging from 0 to 28. The first 3 items measure insomnia severity (difficulties in initiating and maintaining sleep and waking up too early) and the last

4 items assess sleep satisfaction, noticeability of the sleep problem to others, worry about the sleep problem, and sleep problem's interference with daily functioning. The instrument showed acceptable reliability, convergent validity with other subjective and objective sleep measures, and sensitivity to change after treatment (Cronbach $\alpha$ = 0.90) [44,45].

The perception of online learning satisfaction was assessed with a 20-item questionnaire which was used based on sections I to IV of the Dundee ready education environment measure (DREEM), a validated questionnaire designed to measure the educational environment of medical schools and healthcare professionals. There were 3 parts of this questionnaire: (1) online learning and medical education with 8 questions assessing the platforms students got engaged with and the level of interactivity; (2) Student's perceptions of online learning, which were 5-point Likert-type questions, ranging from strongly disagree to strongly agree, adding 2 questions which explored the perceived benefits and barriers of online teaching; and (3) role of online learning in clinical teaching, including academic development factor questions [46]. The specific questions administered in the survey are available as online in Table S1, supplemental appendices.

Additionally, the respondents provided information about the average number of hours spent online for studying, doing job- or internship-related work, watching TV, using social media, watching video classes, reading, doing research or schoolwork, time spent playing video games on a games console, computer, television, tablet, or smartphone, with the responding items ranging from 0–50 min, 50 min–3 h, and >3 h. They were also asked how many hours on average they spend learning every week.

### 2.5. Data Analysis

In line with the explorative nature of our study, we first described the demographics of our participants. The internal consistency of the depression, anxiety, insomnia, and online learning satisfaction scales was assessed using Cronbach's alpha coefficient and all instruments had coefficients above 0.7 which is considered to be the minimum acceptable value for proper reliability [47]. We then correlated the main variables of this study, namely depression, anxiety, insomnia, and online learning perceived satisfaction, with each other. Thirdly, we used multiple regression with perceived satisfaction with online learning as the dependent variable and insomnia, depression, anxiety, and average number of hours studying as predictors in order to see which predictor is related to satisfaction unaffected by the others, how much bigger the level of association is, the direction of association (positive or negative), or even the existence of the association. Fourthly, we ran a separate hierarchical multiple regression with anxiety, insomnia, and depression as dependent variables. We introduced in model 1 the other 2 clinical variables, and in model 2, satisfaction and the average number of hours spent weekly on online learning were used as variables. This was undertaken in order to predict the targeted variable and to determine whether the satisfaction and the average number of hours spent weekly on online learning explain the satisfaction variance in addition to depression and anxiety. Lastly, in the groups of students who answered questions regarding interactivity, we ran a series of ANOVA and t-tests in order to assess the differences in the dependent variables. We opted for a Sidak post hoc test when needed. The alpha level was 0.05 for all tests. All analyses were conducted with SPSS version 23.0.

## 3. Results

### 3.1. Sociodemographic Data

The response rate was 30.02% (463 out of 1529 medical students invited). Data were collected from 463 students who agreed to participate in the study, but 64 participants were excluded from further analyses due to incorrect answers in the attention-check item, which proved their data were not reliable. As such, the sample contained the remaining 399 participants (133 men, 285 women, 1 not specified gender; mean age = 20.02, SD = 4.047). A total of 79.4% of the participants reported over three hours of online studying, watching

video classes, reading, doing research or schoolwork on a computer, television, tablet, or smartphone, with 77.9% reported spending on average between 0–50 min playing video games and 48.6% spending between one to three hours on social media each day (results in Table S1. Supplementary Materials). Table 1 summarizes the descriptive indicators of the sociodemographic and the main variables of interest.

**Table 1.** Sociodemographic characteristics of the sample.

| Variable | Value |
|:---:|:---:|
| **Age** | 20.02 (4.04) |
| **Gender** | *n* (%) |
| Male | 113 (28.3%) |
| Female | 285 (71.4%) |
| Other | 1 (0.3%) |
| **Language of teaching** | *n* (%) |
| Romanian module | 324 (81.2%) |
| English module | 75 (18.8%) |
| **Main variables of interest** | M (SD) |
| PHQ-9 | 14.68 (7.64) |
| GAD-7 | 12.41 (5.74) |
| ISI | 12.22 (6.56) |
| DREEM | 25.43 (8.18) |
| Average hours studying per week | 28.09 (16.1) |

*3.2. Associations between Depression, Anxiety, Insomnia, and Average Number of Hours Studying (Predictors) with Online Learning Satisfaction (Criterion)*

The results from the regression analysis are outlined in Tables 2 and 3. We identified significant negative correlations between online learning satisfaction and depression, $r(397) = -0.382$, $p < 0.001$, anxiety $r(397) = -0.295$, $p < 0.001$, and insomnia $r(397) = -0.247$, $p < 0.001$, as seen in Table 3, and the average number of hours studying was positively correlated to online learning satisfaction $r(322) = 0.119$, $p = 0.32$. There were confirmed correlations between depression, anxiety, and insomnia ($p < 0.05$). Depression, anxiety, and insomnia are not correlated with the average number of hours studying. The satisfaction with online learning is significantly predicted, $R^2 = 0.12$, $F(4, 319) = 11.06$, $p < 0.001$, negatively by the depression scores and positively by the number of hours studying per week (see Table 2).

**Table 2.** Correlations between main variables of interest.

| Variables | n | 1 | 2 | 3 | 4 | 5 |
|:---:|:---:|:---:|:---:|:---:|:---:|:---:|
| 1. DREEM | 399 | — | | | | |
| 2. GAD-7 | 399 | −0295 ** | — | | | |
| 3. PHQ-9 | 399 | −0382 ** | 0.745 ** | — | | |
| 4. ISI | 399 | −0247 ** | 0.609 ** | 0.711 ** | — | |
| 5. Average hours per week | 324 | 0.119 * | 0.038 | 0.082 | 0.054 | — |

Note. GAD-7: generalized anxiety disorder-7. PHQ-9: the patient health questionnaire. ISI: insomnia severity index. DREEM: Dundee ready education environment. * $p < 0.05$. ** $p < 0.01$.

**Table 3.** Multiple linear regression predicting online learning satisfaction.

| Variable | B | SE | 95% CI | | β | t | p |
|---|---|---|---|---|---|---|---|
| | | | LL | UL | | | |
| GAD-7 | −0.061 | 0.111 | −0.278 | 0.157 | −0.042 | 0.58 | 0.582 |
| PHQ-9 | −0.351 | 0.095 | −0.538 | −0.165 | −0.326 | 3.71 | 0.001 ** |
| ISI | 0.053 | 0.096 | −0.135 | 0.241 | 0.042 | 0.55 | 0.579 |
| Average hours studying per week | 0.072 | 0.026 | 0.021 | 0.123 | 0.145 | 2.76 | 0.006 ** |

Note. GAD-7: generalized anxiety disorder-7. PHQ-9: the patient health questionnaire. ISI: insomnia severity index. DREEM: Dundee ready education environment. ** $p < 0.01$.

### 3.3. Satisfaction with Online Learning and Average Number of Hours Studying per Week as Predictors of Psychopathological Variables

Considering that these constructs are highly correlated [48], we built one hierarchical multiple regression model for every variable. Each regression followed the following procedure: in step 1 we introduced only the two other clinical variables than the criterion variable, and in step 2 we introduced online learning satisfaction and average hours spent studying per week. In this way, the high correlations are statistically controlled. The average number of hours spent studying and satisfaction do not predict insomnia and anxiety when we control statistically for depression/anxiety and depression/insomnia. However, satisfaction and the average number of hours studying explain together depression scores significantly more than anxiety and insomnia alone, $R^2$ change = 0.017, $F(2, 319) = 152.551$, $p = 0.001$. Moreover, in this model, satisfaction is negatively associated with the level of depression, B = −0.118, t = −3.713, $p < 0.001$, but not the average number of hours studying, B: −0.11, t = 1.75, $p = 0.081$, as shown in Table 4.

**Table 4.** Hierarchical multiple regression: satisfaction and hours spent studying predicting depression.

| Predictor Variables | ꭕ$R^2$ | $R^2$ | B | SE | 95% CI | | β | t | p |
|---|---|---|---|---|---|---|---|---|---|
| | | | | | LL | LL | | | |
| Step 1 | 0.64 | 0.64 | | | | | | | <0.001 *** |
| GAD-7 | | | 0.62 | 0.05 | 0.51 | 0.73 | 0.47 | 11.32 | <0.001 *** |
| ISI | | | 0.5 | 0.04 | 0.4 | 0.6 | 0.42 | 10.26 | <0.001 *** |
| Step 2 | 0.01 | 0.65 | | | | | | | 0.001 *** |
| DREEM | | | −0.11 | 0.03 | −0.18 | −0.05 | −0.12 | −3.71 | <0.001 *** |
| Average hours studying per week | | | 0.02 | 0.01 | −0.003 | 0.05 | 0.05 | 1.75 | 0.81 |

Note. GAD-7: Generalized anxiety disorder-7. PHQ-9: the patient health questionnaire. ISI: insomnia severity index. DREEM: Dundee ready education environment. *** $p < 0.001$.

### 3.4. Differences in Psychopathology and Satisfaction Depending on Interactivity, Academic Development Factors, and Communication Tools in Online Teaching

We identified through several one-way ANOVAs that there are significant differences between students grouped by their rating of interactivity in satisfaction with online learning scores, $F(3, 395) = 51.87$, $p < 0.001$, $\eta^2 = 0.284$, anxiety, $F(3, 395) = 4.244$, $p = 0.006$, $\eta^2 = 0.031$, depression, $F(3, 395) = 6.630$, $p < 0.001$, $\eta^2 = 0.048$, and insomnia $F(3, 395) = 3.432$, $p = 0.017$, $\eta^2 = 0.026$.

Furthermore, we ran multiple post hoc analyses using the Sidak post hoc criterion for significance which revealed a complex pattern of pairwise significant differences on every mentioned dependent variable (see Table 5).

**Table 5.** Pairwise comparisons between reported interactivity levels on main variables of interest.

| | "Are These Online Teaching Sessions Interactive?" | | | | | | | | | | | | | | |
| | Responses [a] | | | | Pairwise Comparisons: Sidak Post Hoc Test and Cohen's d | | | | | | | | | | |
| | 4 | 3 | 2 | 1 | 4-3 | | 4-2 | | 4-1 | | 3-2 | | 3-1 | | 2-1 |
| | n = 59 | n = 116 | n = 180 | n = 41 | | | | | | | | | | | |
| | M (SD) | M (SD) | M (SD) | M (SD) | $M_4-M_3$ | d | $M_4-M_2$ | d | $M_4-M_1$ | d | $M_3-M_2$ | d | $M_3-M_1$ | d | $M_2-M_1$ |
| GAD-7 | 10.54 (6.09) | 11.88 (5.21) | 12.90 (5.66) | 14.17 (6.2) | −1.34 | | −2.36 * | 0.41 | −3.63 * | 0.59 | −1.02 | | −2.29 | | −1.27 |
| PHQ-9 | 11.83 (7.81) | 13.87 (6.84) | 15.27 (7.48) | 18.14 (8.54) | −2.05 | | −3.45 * | 0.46 | −6.32 * | 0.78 | −1.4 | | −4.27 * | 0.58 | −2.87 |
| ISI | 11 (7.48) | 11.86 (6) | 12.2 (6.29) | 15.07 (7.30) | −0.86 | | −1.2 | | −4.07 * | 0.55 | −0.34 | | −3.21 * | 0.5 | −2.87 |
| DREEM | 34.11 (7.69) | 27.44 (7.42) | 22.68 (6.13) | 19.92 (7.77) | 6.67 * | 0.89 | 11.43 * | 1.75 | 14.19* | 1.84 | 4.75 * | 0.71 | 7.51 * | 1 | 2.76 |

Note. GAD-7: generalized anxiety disorder-7. PHQ-9: the patient health questionnaire. ISI: insomnia severity index. DREEM: Dundee ready education environment. [a] the response categories are: 4 = "Yes", 3 = "Majority are", 2 = "Majority are not", 1 = "No", * $p < 0.05$.

Academic development factors measured in this study were (1) the extent to which online learning successfully replaced its on-site version and (2) the extent to which medical students feel they can learn practical professional skills. As such, we first analyzed the data from the item "Does the online learning replace the physical one?" at which students responded with "Yes," "Yes to some extent," and "No". We ran a series of one-way ANOVAs which detected significant differences among the response groups in satisfaction, $F(2, 373) = 87.72$, $p < 0.001$, $\eta^2 = 0.32$, depression, $F(2, 373) = 10$, $p < 0.001$, $\eta^2 = 0.052$, and insomnia, $F(2, 373) = 4.18$, $p = 0.016$, $\eta^2 = 0.022$, but not anxiety, $p = 0.08$. We ran several Sidak post hoc tests which revealed a pattern of significant pairwise differences summarized in Table 6.

**Table 6.** Pairwise comparisons between levels of reported replacement of on-site learning on main variables of interest.

| | **"Does the Online Learning Replace the Physical One?"** | | | | | | | | |
|---|---|---|---|---|---|---|---|---|---|
| | **Responses** | | | **Pairwise Comparisons: Sidak Post Hoc Test and Cohen's d** | | | | | |
| | **3** | **2** | **1** | **3-2** | | **3-1** | | **2-1** | |
| | **n = 22** | **n = 92** | **n = 260** | | | | | | |
| | **M (SD)** | **M (SD)** | **M (SD)** | $M_3 - M_2$ | **d** | $M_3 - M_1$ | **d** | $M_2 - M_1$ | **d** |
| GAD-7 | 11.5 (6.9) | 11.32 (5.86) | 12.81 (5.64) | 0.17 | | −1.32 | | −1.48 | |
| PHQ-9 | 13.36 (9.72) | 11.64 (7.31) | 15.72 (7.46) | 1.72 | | −2.36 | | −4.08 * | 0.55 |
| ISI | 11.45 (8.85) | 10.56 (5.84) | 12.81 (6.53) | 0.89 | | −1.36 | | −2.25 * | 0.35 |
| DREEM | 36.81 (8.81) | 31.13 (7.44) | 22.38 (6.44) | 5.68 * | 0.74 | 14.43 * | 2.17 | 8.74 * | 1.3 |

Note. GAD-7: generalized anxiety disorder-7. PHQ-9: the patient health questionnaire. ISI: insomnia severity index. DREEM: Dundee ready education environment. the response categories are: 3 = "Yes", 2 = "Yes, to some extent", 1 = "No"; * $p < 0.05$.

The other academic development factor was measured through the question "Do you feel able to learn practical clinical skills through online learning?" to which students responded with "Yes," "Yes, to some extent," and "No". We analyzed the data with a one-way ANOVA for every main variable and we obtained significant differences in satisfaction, $F(2, 378) = 48.76$, $p < 0.001$, $\eta^2 = 0.2$, but not in anxiety, depression, or insomnia.

*3.5. Pleasant Aspects and Barriers Perceived in Online Learning Satisfaction and Their Relationship with Psychopathology*

We studied which aspects of online learning help interactivity related to anxiety, depression, and insomnia. We analyzed aspects such as interactions through chatbox, live speech, and live quiz, among others. The t-tests for the aspect "chatbox" showed us that participants who marked it had better online learning satisfaction scores (t(df) = 397, $p < 0.001$) but were not related to the results on anxiety, depression, and insomnia. Furthermore, the opportunity to learn through live speech (t(df) = 397, $p = 0.53$), live quiz (t(df) = 397, $p = 0.53$), or other methods (t(df) = 397, $p = 0.31$) showed no differences in any of the dependent variables (the data from these analyses are attached in Supplementary Table S1).

Furthermore, we tried to highlight the pleasant aspects of online learning to understand their relationship with satisfaction and psychopathological variables. We analyzed through several t-tests aspects such as "no travel," "saving up," "possibility of asking questions," "self-paced learning," and "flexibility" (data available in Table S1).

There were significant differences in online learning satisfaction depending on the absence or presence of the following pleasant aspects: "no travel," "saving up," "possibility of asking questions," and "own pace." Those who checked "possibility of asking questions," "own pace," and "flexibility" had significant differences in insomnia. Regarding depression, there were significant differences depending on "possibility of asking questions," "own pace," "more comfortable," "flexibility," and "other reasons." Furthermore, those

who reported "possibility of asking questions" and "own pace" as pleasant aspects had significantly lower results in anxiety.

We also assessed the negative aspects (barriers perceived) in online learning satisfaction through a series of *t*-tests. We found a complex pattern of differences in the investigated psychopathological variables. There were significant differences in insomnia depending on the presence or absence of the following barriers: "internet connection," "courses schedule," "family distractions," and "lack of technology," but not "lack of space." Regarding depression, there were significant differences depending on "courses schedule" and "lack of space" but not "internet connection" or "family distractions." Furthermore, the participants who reported "courses schedule," "family distractions," and "lack of space" as barriers also reported higher levels of anxiety compared with those who did not. However, the presence of each barrier significantly lowers the online learning satisfaction, except for "internet connection" (data available in Table S1).

## 4. Discussion

Online learning was faced with many challenges. Firstly, the lack of infrastructure, lack of hardware (such as tablets and laptops for teachers and students), and lack of pre-existing online learning platforms were significant challenges. Secondly, the availability of materials and technology for online learning was limited. Third, limited social interaction between teachers and students and between student communities led to an unprecedented situation, with the psychological impact and associated individual psychiatric symptoms being some of the most studied in the literature [49]. Despite this, Romania adapted to cope with the crisis. It moved most courses online and delivered lectures through digital platforms.

To the best of our knowledge, this is the first Romanian study to provide information on medical undergraduates' self-reported psychopathology and its relationship with online learning satisfaction after the social isolation period. This study aimed to offer an overall perspective on medical students' perception of online learning satisfaction and its relation to self-reported psychopathology and the average hours spent studying online. Additionally, we explored the interactivity factor and its relationship with psychopathology. Furthermore, we studied the impact of the pleasant aspects and the perceived barriers of online learning on satisfaction and psychopathology, if any. Throughout the study, the methodology paid special attention to the rationale of psychological instruments used, the accuracy of the data, of making the participants aware of the cause and potential effects of the study in benefiting future awareness and research in this area, and also the privacy policy.

### 4.1. Associations between Depression, Anxiety, Insomnia, and Average Number of Hours Studying with Online Learning Satisfaction

Firstly, we identified high scores of self-reported depression, anxiety, and insomnia with mean scores highlighting moderately severe depression, moderate anxiety, and subthreshold insomnia with the average number of hours studying online or spent in total online sky-rocketing in the vast majority of the participants. This is aligned with previous results from a study that assessed psychopathology and average hours spent online in medical undergraduates [50–53].

Depression, anxiety, and insomnia were identified as negatively associated with online learning satisfaction, a finding also reported in previous literature [32,35]. In contrast, the average number of hours studying spent online was identified as a positive predictor. Here, we can emphasize the digital familiarity of the students having a positive contribution to the overall satisfaction, as confirmed in previous findings but without a direct relationship with satisfaction [54]. Furthermore, depression, anxiety, and insomnia were correlated with each other while the average number of hours spent studying online were not correlated with them. A marginal correlation between depression and hours spent studying exists, which may indicate that, even with high depression scores, the students spent more time studying, highlighting an already existing academic pressure on medical students which was amplified during online learning, a topic which has already been discussed in previous studies [54,55].

Moreover, after we performed a multiple regression analysis, we found that only the level of depression and the average number of hours spent studying significantly predicted satisfaction as independent predictors, meaning that the higher the levels of depression and the lower the number of hours studying online, the lower the satisfaction was. This was not the case for anxiety and insomnia.

In order to better understand the potential bidirectional relationship between the variables studied, we found that the average number of hours studying and satisfaction do not predict insomnia and anxiety when we control statistically for depression/anxiety and depression/insomnia. Satisfaction is negatively associated with the level of depression, meaning that the increase in one point on the satisfaction scale leads to the decrease in the level of depression by 0.11; this is not the case with the average number of hours studying when we control statistically for the level of anxiety and insomnia (B = −0.118, t = −3.713, $p < 0.001$). The results were consistent with previous findings regarding depression [56,57] but not with anxiety and insomnia results [58]. We should mention that most of the studies concerning psychopathology related to online satisfaction assessed stress response, burnout symptoms, coping mechanisms, and resilience, not specific psychiatric syndromes [59–61]. Moreover, there is need for further longitudinal studies to investigate if satisfaction with online learning can reduce symptoms of depression as well as if higher depression scores will impact the satisfaction or not.

### 4.2. Interactivity

Regarding interactivity, the majority of the participants reported that most sessions were not interactive. In terms of the relationship with online learning satisfaction, we identified through ANOVA tests and Sidak post hoc tests that in the response groups, "Majority are" and "Yes" compared to "No" interactivity matters more for satisfaction. The higher the perception of positive interactivity, the higher the satisfaction; this result gave a huge size effect (eta squared = 0.28, $p < 0.05$), as seen in a previous study [62]. Furthermore, there were significant differences among ratings of interactivity in anxiety, depression, and insomnia levels with anxiety having the highest impact ($\eta^2 = 0.031$, $p = 0.006$). Furthermore, we studied the aspects which support interactivity and their relationship with psychopathology and satisfaction, and we found that participants who checked "chat box interactions" had better online learning satisfaction scores but were not related to psychopathology; the opportunity to learn through "live speech," "live quiz," or "other reasons" showed no differences in any of the dependent variables. Interactivity was highlighted as a key point in a higher quality of teaching and overall student satisfaction, with student engagement being a key factor in enhancing students' desirable learning outcomes [63,64]. Online learning was not perceived as a better replacement for the traditional one with the satisfaction being impacted in those who responded "No" and "Yes to some extent." These results confirmed some studies even if there were findings where online learning successfully replaced physical learning in terms of efficacy, academic performance, and stress levels [54]. If students did not perceive online learning as a successful replacement for classic learning, the depression and insomnia scores were higher ($\eta^2 = 0.052$; $\eta^2 = 0.022$, $p < 0.001$). This finding highlights the special role of individual psychopathology in evaluating the learning format beyond most of the external variables studied, such as coping mechanisms, academic performance, economic gain, and social isolation. In assessing the perception of how online learning was able to teach practical skills, the majority disagreed with the affirmation with online learning satisfaction being associated with the negative responses, however not with psychopathology.

The novelty our data bring regarding self-reported psychopathology opens a new discussion field regarding the relationship between interactivity and perceived satisfaction beyond previous studied variables including academic performance and self-efficacy [65]. Our findings emphasize an important predictor role of the individual psychopathology which may not have been reported or diagnosed and may mediate in a critical way the perception of the online learning format of teaching. Our results confirmed previous

studies regarding low interactivity within the online learning format [66,67], despite some studies reporting interactivity among the most pleasant aspects [54] with social, economic differences regarding technical possibilities, and various cultural attitudes having an important impact on the results. This study was completed after two years from the onset of the pandemic, and thus a year after many local educational adaptations; however, most students found online learning to be less effective than the classic format, reporting fewer issues with technology but significant individual distress.

### 4.3. Pleasant Aspects and Barriers Perceived in Online Learning and Their Relationship with Psychopathology

In terms of aspects which impacted the perception of online learning satisfaction, we found a statistically significant result for the pleasant aspect of "no travel" meaning that those participants who checked this box scored higher in online satisfaction. Furthermore, "saving up," "possibility of asking questions," and "learning at own pace" had significant differences in all the scales used (Cohen's d-0.46).

Online learning satisfaction scores were higher in those who checked "no travel," "saving up," "possibility of asking questions," "own learning pace," "flexibility," "other reasons," "more comfortable," and "flexibility," and most of these are the same factors checked in previous findings [64,68]. Depression scores were lower in those who checked "possibility of asking questions," "own pace," "more comfortable," and "flexibility." Anxiety scores were lower in those who checked "possibility of asking questions" and "own pace." Insomnia scores were lower in those who checked "possibility of asking questions," "own pace," and "flexibility." Thus, we identified lower scores on all scales of psychopathology for the items that addressed the student's need for interactivity and the ability to learn at their own pace, while items focused on external benefits, such as the time spent on remote or extra financial gain, were not related to psychopathology. This result highlights individual distress but does not show an impact on the relationship or teacher–student connection. Interaction and the ability to ask questions were the least checked by the students, confirming the overall lack of satisfaction with online learning. These findings reflect previous results concerning the perceived benefits of online learning and improved psychopathology related to the perceived pleasant aspects [69].

Regarding the barriers perceived, students had higher insomnia scores when checking "internet connection," "courses schedule," "family distractions," "lack of technology," and "anxiety" (Cohen's d-0.34). Higher scores in depression and anxiety were associated with the items "courses schedule," "family distractions," and "lack of space," while online learning satisfaction was associated with "courses schedule," "family distractions," "lack of space," "lack of technology," and "anxiety." The modified schedule with courses being programmed one after another and family being present most of time while the student was learning were the most frequent items checked in terms of barriers, while the lack of devices was not associated with any of the variables studied. Students found it difficult to be engaged with the lessons with previous barriers being confirmed in a similar study [64].

### 4.4. Limitations

One of the limitations of the present study was that the sample size and response-rate was rather small, and the participation in the survey was based on self-selection. The sample was asymmetric in terms of gender, country of origin, and year of study, although this reflects the real proportion of students in our university. The students included in this study had one of the longest durations of online learning with almost two years experiencing this format, including the last year of their high school program. The design of the study was cross-sectional, thus not allowing the evaluation of the study variables in their short- and long-term dynamics, with the absence of a pre-pandemic dataset being another limitation. This would have provided a greater insight into the relationship satisfaction with online learning and would have offered psychopathology and psychopathology variance before and after the pandemic. Furthermore, because the research was performed just after exam

season, we did not have access to the grades of the students; therefore, we could not assess academic performance in order to measure effectiveness, as in the field of medicine satisfaction is greatly associated with performance and learning. Bias may have played a role as the survey was sent by the faculty members involved in teaching students. The level of digital competence and the ease of online communication were not considered preconditions for the inclusion of participants in the study.

### 4.5. Future Research

Future research should focus on more in-depth exploration of the relationship between satisfaction with online learning and psychopathology, including additional individual (personality factors), demographic, and COVID-19-related variables (e.g., economic difficulties, affected family members, grief, and social isolation), but also screen time usage. This would help researchers to identify potential multiple-factor etiologies (during the pandemic, immediately after, and post-pandemic) and conditions in which psychopathology is present in medical undergraduates, factors which can be quickly addressed within a more systematic, long-term educational and medical approach. Psychopathology and online learning satisfaction could be assessed in relation to academic performance and pressure. The evaluation could not only supplement the current data but also offer an opportunity for implementing individual and group measures for more openness towards and a better quality of online learning while limiting as much as possible the psychopathology burden.

## 5. Conclusions

The present results are significant as they not only confirm previous associations found by various researchers but enable us to explore more in depth the associations between the variables as independent predictors. As the study was conducted after two years of online learning, at the moment of administration the prospect of classical learning was publicly expressed in Romania, and we discovered that depression and online learning satisfaction have a bidirectional relationship with the average number of hours spent studying online as a protective factor in perceived satisfaction but not in psychopathology. Furthermore, anxiety and insomnia negatively predicted online learning satisfaction. Moreover, the results highlighted a key factor in interactivity as it plays a great role in self-reported psychopathology and online learning satisfaction. Perceived benefits concerning individual comfort were associated with lower psychopathology and higher satisfaction and external benefits, such as economic gain and time saving, were not related to psychopathology. The more students believed that online learning did not replace physical learning, the more psychopathology and less satisfaction students reported.

Students need more in-depth psychological assessment and support given the alarming scores on psychopathology Institutions should reflect on how they can ensure, adapt, and provide a higher quality of online learning for better overall satisfaction, while reducing the unpleasant psychological consequences and contributing to individual, community, and academic growth and gratification. With online learning in addition to physical learning having proven benefits in raising motivation, self-efficacy, and academic performance, institutions should consider the psychological well-being of participants, teachers, and students, in future hybrid models of education.

**Supplementary Materials:** The following are available online at https://www.mdpi.com/article/10.3390/ejihpe13030045/s1, Table S1: Supplementary Materials-All the scales used in the study.

**Author Contributions:** Conceptualization, C.G.I.; methodology, C.G.I.; software, C.G.I.; validation, M.L. and A.C.; formal analysis, C.G.I. and A.C.; investigation, C.G.I. and M.L.; resources, C.G.I. and A.C.; data curation, C.G.I. and M.L.; writing—original draft preparation, C.G.I. and A.C.; writing—review and editing, M.L.; visualization, A.C. and M.L.; supervision, C.G.I.; project administration, C.G.I.; funding acquisition, C.G.I. All authors have read and agreed to the published version of the manuscript.

**Funding:** This research received no external funding.

**Institutional Review Board Statement:** The study was conducted according to the guidelines of the Declaration of Helsinki and approved by the Ethics Committee of the University of Medicine and Pharmacy Carol Davila—Bucharest (no. 7872/2021).

**Informed Consent Statement:** Informed consent was obtained from all subjects involved in the study.

**Data Availability Statement:** The data presented in this study are available on reasonable request from the corresponding author.

**Acknowledgments:** The authors would like to thank all the students that participated in the study. A special thanks for the dedication and supportive feedback of Tobias R. Spiller, who helped and inspired us throughout the evolution of this work and of ourselves as researchers.

**Conflicts of Interest:** The authors declare no conflict of interest.

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
