# Peer review of "Is Satisfaction with Online Learning Related to Depression, Anxiety, and Insomnia Symptoms? A Cross-Sectional Study on Medical Undergraduates in Romania"

_ejihpe, doi:10.3390/ejihpe13030045_

Round 1
Reviewer 1 Report
Thank you for the opportunity of reading and reviewing your manuscript. The paper addresses a topic related to online learning and its implications and link with psychopathology. Basically the research is survey based, and the authors present and discuss the results. The paper has merit, but there are several shortcomings that need to be addressed, as follows:
1.there is no clear identified research gap, and the aims should be presented in a more clear manner
2.there is no literature section, there are only several references to some papers. The literature on the topic is quite prolific in the latest years, so that a organized dedicated literature section is needed
3.there are no hypotheses formulated, and I suggest to do this in accordance with the relevant literature
4.there is still the question of the sample and its relevance; please clarify
5.it is necessary to include in the appendix the questionnaire used for investigation.
Good luck!
Author Response
Please see the atachment.

Reviewer 2 Report
Thanks so much for giving me the opportunity to review this paper which is new in the field. Overall good effort considering a cross-sectional study. It will be great if team can work on improving and adding below information
1. Elaborating what classes courses these first-year medical students were enrolled in as first-year classes and curriculum vary across the globe
2. Was the nature of these online classes was like strictly fixed just like in-person classrooms' fixed schedule where they had to log in at certain times with live lectures and active learning where students were supposed to turn on their cameras or were given the option to watch recordings with passive learning?
3. Line 84 needs clarification and grammar correction. Do you mean an average number of hours spent studying online? Or the online time is total screen time including non-educational activities.
4. It will be great if you could mention academic performance in tests/exams to measure effectiveness as in the field of medicine satisfaction is greatly associated with performance and learning.
Round 2
Reviewer 1 Report
Thank you for providing the revised version of your manuscript and for clarifying certain aspects.
Author Response
Dear Reviewer 1,
Thank you again for making a valuable impact with your suggestions on our research and paper helping us improving and adding quality into our study presentation.
With consideration,
Authors